# Incidence, Compliance, and Risk Factor Associated with Central Line-Associated Bloodstream Infection (CLABSI) in Intensive Care Unit (ICU) Patients: A Multicenter Study in an Upper Middle-Income Country

**DOI:** 10.3390/antibiotics14030271

**Published:** 2025-03-07

**Authors:** Arulvani Rajandra, Nor’azim Mohd Yunos, Chin Hai Teo, Anjanna Kukreja, Nur Alwani Suhaimi, Siti Zuhairah Mohd Razali, Sazali Basri, Cindy Shuan Ju Teh, Chee Loon Leong, Ismaliza Ismail, Azureen Azmel, Nor Hafizah Mohd Yunus, Giri Shan Rajahram, Abdul Jabbar Ismail, Shanti Rudra Deva, Pei Wei Kee, TRGS Working Group, Sasheela Sri La Sri Ponnampalavanar

**Affiliations:** 1Department of Medicine, Faculty of Medicine, Universiti Malaya, Kuala Lumpur 50603, Malaysia; arulvani@um.edu.my (A.R.); anjanna@ummc.edu.my (A.K.); sazali@ummc.edu.my (S.B.); 2Department of Anesthesiology, Universiti Malaya Medical Centre, Kuala Lumpur 50603, Malaysia; norazim@um.edu.my; 3Department of Primary Care Medicine, Faculty of Medicine, Universiti Malaya, Kuala Lumpur 50603, Malaysia; teoch@um.edu.my; 4Department of Infection Control, Universiti Malaya Medical Centre, Kuala Lumpur 50603, Malaysia; alwani.s@ummc.edu.my (N.A.S.); zuhairah@ummc.edu.my (S.Z.M.R.); 5Department of Medical Microbiology, Faculty of Medicine, Universiti Malaya, Kuala Lumpur 50603, Malaysia; cindysjteh@um.edu.my; 6Department of Infectious Disease, Hospital Kuala Lumpur (HKL), Jalan Pahang, Wilayah Persekutuan, Kuala Lumpur 50586, Malaysia; bkho@hotmail.com (C.L.L.); ismalizaismail1312@gmail.com (I.I.); 7Department of Infectious Disease, Hospital Tengku Ampuan Rahimah (HTAR), Jalan Langat, Klang 41200, Malaysia; aazmel@gmail.com; 8Department of Anesthesiology, Hospital Tengku Ampuan Rahimah (HTAR), Jalan Langat, Klang 41200, Malaysia; hafizahyunus72@gmail.com; 9Department of Medicine, Hospital Queen Elizabeth (II), Sabah, Lorong Bersatu, Off Jalan Damai, Kota Kinabalu 88300, Malaysia; gsrajahram@gmail.com; 10Department of Anesthesiology and Intensive Care, Faculty of Medicine and Health Sciences, Jalan Universiti Malaysia Sabah (UMS), Kota Kinabalu 88400, Malaysia; abduljabbarismail@ums.edu.my; 11Department of Anesthesiology, Hospital Kuala Lumpur (HKL), Jalan Pahang, Wilayah Persekutuan, Kuala Lumpur 50586, Malaysia; shantiviv@yahoo.com (S.R.D.); janice8@hotmail.com (P.W.K.)

**Keywords:** central line, CLABSI, intensive care unit (ICU), risk factor, microorganism

## Abstract

**Background:** Despite significant prevention efforts, the incidence of central line-associated bloodstream infection (CLABSI) in intensive care units (ICUs) is rising at an alarming rate. CLABSI contributes to increased morbidity, mortality, prolonged hospital stays and elevated healthcare costs. This study aimed to determine the incidence rate of CLABSI, compliance with the central venous catheter (CVC) care bundle and risk factors associated with CLABSI among ICU patients. **Method:** This prospective observational study was conducted in one university hospital and two public hospitals in Malaysia between October 2022 to January 2023. Adult ICU patients (aged > 18 years) with CVC and admitted to the ICU for more than 48 h were included in this study. Data collected included patient demographics, clinical diagnosis, CVC details, compliance with CVC care bundle and microbiological results. All data analyses were performed using SPSS version 23. **Results:** A total of 862 patients with 997 CVCs met the inclusion criteria, contributing to 4330 central line (CL) days and 18 CLABSI cases. The overall incidence rate of CLABSI was 4.16 per 1000 CL days. The average of overall compliance with CVC care bundle components was 65%. The predominant causative microorganisms isolated from CLABSI episodes were Gram-negative bacteria (78.3%), followed by Gram-positive bacteria (17.4%) and *Candida* spp. (2.0%). Multivariate analysis identified prolonged ICU stay (adjusted odds ratio (AOR): 1.994; 95% confidence interval (CI): 1.092–3.009), undergoing surgery (AOR: 2.02, 95% CI: 1.468–5.830) and having had multiple catheters (AOR: 3.167, 95% CI: 1.519–9.313) as significant risk factors for CLABSI. **Conclusions:** The findings underscore the importance of robust surveillance, embedded infection-control and -prevention initiatives, and strict adherence to the CVC care bundle to prevent CLABSI in ICUs. Targeted interventions addressing identified risk factors are crucial to improve patient outcomes and reduce healthcare costs.

## 1. Introduction

Central venous catheters (CVCs) are vital for managing critically ill patients in intensive care units (ICUs). They are used for various procedures, including hemodialysis, parenteral nutrition, chemotherapy, and monitoring [1,2]. However, while CVCs provide significant benefits, they also pose risks, including mechanical injuries and nosocomial infections like bloodstream infections, septic thrombophlebitis, and endocarditis [3,4]. The most serious complication associated with CVCs is central line-associated bloodstream infection (CLABSI) [2,5,6].

The prevalence of CLABSI is particularly high in resource-limited settings, such as upper middle-income countries, where healthcare infrastructure and infection-prevention practices vary [7,8]. In Asian countries, CLABSI rates range from 1.5 to 17.04 per 1000 catheter days, significantly higher than the 0.5 to 2.2 per 1000 catheter days reported in high-income nations [9,10,11,12,13,14,15,16,17]. In Malaysia, the Malaysian Registry of Intensive Care Unit (MRIC) reported a central venous catheter-related bloodstream infection (CVC-BSI) rate of 0.3 per 1000 catheter days in 2022 [18], while other studies showed rates between 7.83 and 9.4 per 1000 catheter days [19,20,21]. However, it is important to note that CVC-BSI and CLABSI rates are not directly comparable due to different diagnostic criteria. This highlights the variability and underscores the need for standardized definitions and reporting methods. The impact of CLABSI includes increased mortality, higher hospitalization costs, and prolonged hospital stays [1,22]. In addition, recent studies, especially from Asian countries, have highlighted the growing concern surrounding Gram-negative multidrug-resistant organisms (MDROs) as the predominant pathogens in CLABSI [10,12,13,23]. The emergence of MDROs as causative pathogens increases the risk of poorer outcomes due to their resistance to standard treatments, complicating effective infection management [24,25].

CLABSI can occur through two main mechanisms: endogenous and exogenous transmission [26]. Endogenous transmission happens when microorganisms already present in the patient enter the bloodstream, often during catheter insertion or when accessing the catheter hub [13,27]. Exogenous transmission occurs when foreign microorganisms enter the body from the environment, frequently due to the contaminated hands of healthcare workers or contaminated infusates [28]. Additionally, CLABSI can result from contamination of the catheter hub, particularly after 10 days, often due to lapses in aseptic techniques [29]. This highlights the importance of strict aseptic techniques during catheter insertion and maintenance in preventing such contamination and infection. Understanding the risk factors and practices associated with CLABSI is crucial for reducing morbidity and mortality associated with this infection.

Preventing CLABSI is a major focus of infection-control strategies worldwide, employing approaches such as compliance with care bundles, education and monitoring [6,15,30,31]. Various international guidelines recommend evidenced-based strategies to reduce CLABSI rates [31,32,33,34]. These strategies are often implemented through a care bundle approach, which consists of packages of at least three to five evidence-based strategies grouped together to maximize their effectiveness and optimize patient care [35]. Care bundles are widely used in healthcare settings to prevent and manage various health conditions such as in preventing ventilator use associated with pneumonia, surgical site infection (SSI) and pressure ulcer [35,36]. Insertion and maintenance of central venous access device bundles demonstrate reductions in the frequency of complication and bloodstream infection when implemented with compliance monitoring [37,38,39]. In Malaysia, the Ministry of Health (MOH) introduced a CVC care bundle guideline which included a checklist in 2016 aligning local practices with international standards and to address this ongoing challenge [40,41]. The components of the CVC care bundle include the following practices: (1) hand hygiene, (2) maximal barrier precautions, (3) chlorhexidine skin antisepsis, (4) optimal site selection and (5) daily review of (CVC) line [40]. Despite these efforts, compliance with care bundles remains unknown, and the CLABSI rates continue to increase (Ministry of Health, Malaysia data 2020). Understanding the risk factors associated with CLABSI is crucial for developing targeted interventions for prevention and management. Therefore, this study aims to determine the prevalence of CLABSI, identify its associated risk factors, and assess the CVC care bundle compliance in the ICUs in Malaysia.

## 2. Results

### 2.1. Patients’ Demographic Data and Central Line (CL)-Related Details

A total of 862 patients were admitted to the ICUs, with 997 CVCs fulfilling the inclusion criteria during the study period. The surveillance data reported 4330 central line days and 18 CLABSI cases. Table 1 shows the characteristics of patients. The median age of ICU patients was 59.0 and 60.0% were males (n = 517). The median length of stay in the hospital for the entire patient cohort was 15 days and the duration of ICU hospitalization was 8 days. About 37.0% of patients had undergone surgery. The overall median of APACHE II score was 19.0 and the Charlson Comorbidity Index was 4.0, respectively. The overall ICU mortality rate was approximately 20%.

Data related to the CLs, including the site of insertion, number of lumens and use of lock solution are presented in Table 1. The femoral vein (57.9%) was the most frequently used insertion site, followed by the jugular vein (39.3%). The majority of central lines had triple lumens (86%); the median days between line insertion and CLABSI diagnosis was 6 days.

### 2.2. CLABSI Rates and Compliance with CVC Care Bundle

The overall incidence rate of CLABSI was 4.16 per 1000 CL days. The CLABSI rate per 1000 CL days in four ICUs from October 2022 to January 2023 was 3.38, 4.27, 3.78 and 5.21 per 1000 CL days, respectively (Figure 1). The overall compliance with CVC care bundle components from October 2022 to January 2023 was 64.91%, 68.25%, 65.85% and 62.82%, respectively. Based on the data in Appendix A, adherence rates for four components of the CVC care bundle were as follows: hand hygiene (69.5%), maximal barrier precautions (69%), chlorhexidine antisepsis (66.9%) and daily review (77.1%). Among these, chlorhexidine antisepsis had the lowest adherence.

### 2.3. Microbiology-Related Data of CLABSI Patients

Table 2 presents the distribution of microorganisms among CLABSI patients in ICU. There were a total of 23 isolates responsible for CLABSI in this study; the majority (22 isolates (98.0%)) were bacterial (18 Gram-negative and 4 Gram-positive) and one (2.0%) *Candida* spp. Twelve of the 23 (52.2%) organisms were multidrug-resistant Gram-negative isolates. The most commonly identified pathogen was Klebsiella pneumoniae with seven isolates (30%), followed by Acinetobacter baumanii, Burkholderia cenocepacia and Stenotrophomonas maltophilia, each with two isolates.

### 2.4. CLABSI Risk Factors

A univariate analysis of the whole cohort before and after propensity score (PS) is illustrated in Table 3. In univariate analysis, duration of ICU stays, patients undergoing dialysis, patients having hypertension, dyslipidemia, a higher CCI, a higher APACHE II score, patients who had more than one CVC in situ, elective and emergency insertion, compliance with hand hygiene, maximal barrier precaution and chlorhexidine antisepsis were factors associated with CLABSI. In the multivariate analysis of the whole cohort before PS matching, only APACHE II score (AOR: 1.052, Cl: 1.006–1.100, *p* = 0.025) and more than one CVC (AOR: 3.401, Cl: 1.024–11.180, *p* = 0.044) were significant risk factors for CLABSI. After PS matching, the univariate analysis showed patients who had had a longer ICU stay, were on dialysis, had had surgery, have hypertension, have dyslipidemia, have a higher CCI or have more than one CVC in situ and also jugular catheter site, category of the operator (medical officer) and use of ultrasound-guided catheter insertion were factors for CLABSI. The results of multivariate binary logistic regression analysis of factors associated with CLABSI after PS matching with adjusted odds ratio (AOR) with 95% confidence interval (CI) were as follows: longer ICU stay patients, 1.994 (1.092–3.009), were significantly more likely to acquire CLABSI. ICU patients who underwent surgery were two times more likely to have CLABSI, 2.021 (1.468–5.830). Patients with more than one catheter in situ were three times more likely to suffer CLABSI infection, 3.167 (1.519–9.313).

## 3. Methodology

### 3.1. Study Design, Setting and Period of Study

A prospective observational study was conducted in one university hospital and two public hospitals in Malaysia between October 2022 and January 2023. The characteristics of the hospitals are described in Table 4.

### 3.2. Inclusion and Exclusion Criteria

All eligible patients who were admitted to an ICU during the study period were recruited. The inclusion criteria were (i) patients who were 18 years old and above and had a central venous catheter, (ii) admitted to the ICU more than 48 h during the study period. The exclusion criteria were patients who died and were discharged from ICU within 48 h of ICU admission. All patients were followed up on until they were transferred out, were discharged from the ICU or had died in the ICU.

### 3.3. Outcomes Measured

The primary outcome was to assess the incidence rate of CLABSI per 1000 central line (CL) days and the rate of compliance with the CVC care bundle. The secondary outcome was to assess the risk factors associated with CLABSI. The following definitions in this study are based on CDC and MOH criteria [40,41], shown in Table 5.

### 3.4. Research Instrument

A standardized case report form Appendix A was adapted and modified based on the Ministry of Health Malaysia BSI surveillance form 2016. The information collected included the patients’ demographic characteristics (age, gender), clinical diagnosis, Charlson Comorbidity Index (CCI) and illness severity, which was evaluated using the Acute Physiology and Chronic Health Evaluation (APACHE) II score during the first 24 h upon ICU admission and comorbidities. In addition, invasive procedures, including treatment such as dialysis, surgery and ambulatory oncology therapy, as well as the presence of other devices in situ such as arterial line, peripheral line, mechanical ventilation (MV) and urinary catheters (CBD/Foley) were also captured. The duration of stay in hospital and ICU were also recorded. CVC information obtained included types, site of insertion, number of lumens, use of lock solution, date of insertion and removal, reason for removal (infection, no longer indicated, malfunction) and documentation of compliance with CVC care bundle (insertion and maintenance)-based MOH guidelines [40]. The CVC care bundle guidelines by the Ministry of Health (MOH) Malaysia were published in 2016. All healthcare facilities are required to comply with these guidelines as part of a national initiative to standardize the use of CVCs in order to reduce CLABSI [40]. The components of the CVC care bundle include the following practices: (1) hand hygiene; (2) maximal barrier precautions; (3) chlorhexidine skin antisepsis; (4) optimal site selection; and (5) daily review of the CVC line. Full compliance was defined as completing all five components of the bundle, and a score of 1 was assigned when each component was met. If any component was not completed, a score of 0 was recorded. The microbiological information collected included the name of organisms, the resistance profile and the origin of infection or colonization.

### 3.5. Data Collection

The ICU infection-control liaison nurse collected data on all inpatients who had CVC in situ using the surveillance form. Daily surveillance was performed to record the number of patients with CVC every day at 8.00 am to determine catheter days. Infection-control nurses collected all positive bloodstream infections in the ICU every month. All the cases were assessed based on the clinical and microbiological criteria for diagnosing CLABSI by an infectious disease physician or infection prevention and control physician. If the team could not reach a consensus regarding the care, a more senior infectious disease physician was consulted. Prior to data entry, the primary investigators and two research assistants from each participating hospital were trained in data extraction by the research coordinator using a standardized training module based on the surveillance protocol. This training included CLABSI identification and definition, as well as compliance assessment with the CVC care bundle to ensure consistency and accuracy in data collection across all study sites. All data gathered were inserted in the Research Electronic Data Capture (REDCap) platform.

### 3.6. Statistical Analysis

Data analysis was performed using Statistical Package for Social Sciences (SPSS) version 23.0 (SPSS; Chicago, IL, USA). Continuous variables were reported in terms of mean ± standard deviation or median with interquartile range, based on normality assessments. CLABSI risk factor analysis was performed in the univariate logistic regression analysis and variables with *p* < 0.25 were then included in the subsequent analysis using multivariate logistic regression to produce an adjusted odds ratio (AOR). A 1:4 propensity score (PS) matching between cohorts with and without CLABSI was applied to determine demographics, catheter type, CVC care bundle (CCB) compliance, and mortality to derive the predicted probabilities. The same analysis of univariate logistic regression analysis and multivariate logistic regression was performed after PS matching.

## 4. Discussion

The rate of CLABSI was 4.16 per 1000 central line (CL) days in the general ICUs of three tertiary referral and one teaching hospital in Malaysia. This rate is slightly lower than the 5.08 per 1000 CL days reported across ICUs in Asia. The previous study also highlighted that patients in lower middle-income countries had nearly twice the likelihood of acquiring a CLABSI (adjusted odds ratio (AOR) 1.87; 95% CI 1.41–2.47) [43]. Similarly, a systematic review of the burden of healthcare-associated infection (HAI) in Southeast Asia, including countries like Indonesia, Malaysia, Philippines, Singapore, Thailand and Vietnam, reported a pooled CLABSI incidence rate of 4.7/1000 CL days [44]. Of note, the CLABSI rate observed in our study was significantly lower compared to earlier studies conducted in Malaysian ICUs. During the period from 2009 to 2015, the reported rates were nearly 10 per 1000 CL days [20,45]. This decrease reflects the cumulative impact of ongoing infection-prevention strategies implemented in recent years, such as the introduction of CVC care bundle practices in 2016 [40], improved knowledge and enhanced training, aligned with Malaysia’s National Patient Safety Goals (Malaysian Action Plan on Antimicrobial Resistance (MyAP-AMR) 2017–2021). In contrast, the observed CLABSI remains significantly higher compared to those reported in developed countries. For instance, the rate per 1000 CL days in the United States was 0.8 [29], Germany 1.5, Switzerland 1.69, Italy 2.0, France 1.23 [46], Australia 1.26 [47] and Poland 1.83 [48]. The observed differences in rate may be attributed to the robust surveillance system in developed countries, which provide accurate epidemiological estimates and better reflect the variability in practices to control HAI [46]. Additionally, successful infection-prevention practices such as the national hand hygiene program, ICU quality-improvement project, and increased staffing may also contribute to their lower rates [47].

The average adherence to completing all three components of the CVC care bundle was approximately 65%. This is lower than that reported by the Malaysia Intensive Care Registry (MRIC, Malaysia), which was 98% [18]. The compliance rates reported from other countries varied from as low as 28.5% in Korea [49] to as high as 95% in the USA [50]. Consistent adherence to CVC care bundles has been shown to decrease CLABSI. A Thai hospital reported ≥75% compliance with all the CLABSI-prevention bundle components, which was significantly associated with a 38.3% (*p* < 0.001) reduction in the CLABSI rate [51]. Similarly, Indian ICUs reported a reduction in CLABSI rate from 3.1 to 0.4/1000 CL days following the implementation of a preventive care bundle [52]. A meta-analysis published in *The Lancet* found that implementing insertion and maintenance bundles significantly reduced CLABSI rates across all age groups, from neonatal to adult, regardless of the country of origin. In the adult ICUs, the reduction ranged from 1.2 to 46.3 per 1000 CL days to 0 to 19.5 per 1000 CL days before and after the adherence improvement, respectively [38]. In addition to reducing CLABSI incidence, care bundles are shown to shorten hospital stay and improve patient well-being [8]. For instance, Sun et al. [8] from Beijing found that the average length of stay decreased from 16.7 days in those receiving conventional nursing care to 12.6 days in those utilizing the CVC care bundle. Furthermore, patients in the care bundle group showed improved psychological well-being, with significantly lower scores on the Self-Rating Anxiety Scale (SAS: 44.3, t = 8.093, *p* < 0.05) and Self-Rating Depression Scale (SDS: 46.2. t = 8.892, *p* < 0.05) [8]. This evidence highlights the importance of implementing all components of the CVC care bundle consistently without exceptions [50,52]. Hence, potential interventions such as standardized training, routine audits and feedback mechanisms to reinforce adherence are essential to sustaining high compliance rates and effectively reducing CLABSI incidence.

In this study, patients with longer stays in the ICU, who had undergone surgery and had multiple catheters in situ, were more likely to develop CLABSI. The presence of multiple lines increases the risk of infection due to frequent manipulations, which may increase the risk of exposure to external factors such as improper disinfection of hub, contamination from skin flora and translocation of pathogens from other body sites [53,54,55]. A retrospective propensity score-matched cohort study conducted in Georgia found that the concurrent use of a second CVC was associated with a twofold higher likelihood of developing CLABSI (adjusted risk ratio, 1.62; 95% CI, 1.10–2.33; *p* = 0.001) [54]. These findings underscore the importance of implementing practical, evidence-based interventions such as removing unnecessary lines to reduce patient harm, as minimizing unnecessary catheter manipulations can significantly lower CLABSI risk and improve patient outcomes [53,54].

Length of stay (LOS) has consistently been identified as a significant risk factor for CLABSI in regional and global studies [20,43,44,45,48,55,56]. In this study, patients with longer ICU stays tended to have almost two times higher rates of CLABSI. A multinational study across nine Asian countries demonstrated that prolonged ICU stays contribute to a 4% daily increase in the risk of CLABSI [43]. Similarly, a study from a Japanese university hospital reported a significant association between a long ICU stay of up to 62 days (OR: 1.032; CI: 1.019–1.044) increases the risk of CLABSI [57]. These findings highlight the importance of minimizing unnecessary catheter days and optimizing ICU discharge processes to reduce CLABSI risk. Strategies such as daily catheter assessment and timely removal of unnecessary lines are crucial to mitigate this preventable infection burden and are key components of the CVC care bundle [41,58].

Invasive procedures have also been identified as a notable risk factor for CLABSI. A systematic review encompassing studies from the USA, South Korea, Australia, Brazil, Egypt, Germany, Spain and Thailand from 2010 to 2023 identified surgery as a significant risk factor of CLABSI (adjusted OR: 3.793; 95% CI: 1.467–9.805; *p* = 0.006) [59]. Notably, patients undergoing emergency surgeries were associated with a twofold increase in CLABSI (OR: 192.1.92, 95% CI 1.02–3.61) [60]. The increased risk of CLABSI in surgical patients can be attributed to several factors, including the increased manipulation of catheters during surgery, challenges in maintaining a sterile environment, extended ICU stays, surgical complexity, the use of multiple lumen catheters and the administration of parenteral nutrition [59]. These findings highlight the importance of implementing targeted CLABSI-prevention strategies, particularly in high-risk surgical settings and postoperative care.

Our data showed that the pathogenic microorganisms associated with CLABSI were predominantly Gram-negative bacteria (78.3%), with Klebsiella pneumoniae being the most commonly isolated pathogen. Half of the isolates were MDRO, all of which were Gram-negative. These findings reflect the regional predominance of Gram-negative bacteria such as Pseudomonas aeruginosa, Klebsiella species and Acinetobacter baumannii, which are frequently implicated in HAIs including CLABSI [10,12,23,44,45]. In contrast, the USA reported higher proportions of Gram-positive organisms, such as methicillin-resistant Staphylococcus aureus (MRSA), where the sources were most likely due to skin commensals [24,29]. The predominance of Gram-negative organisms as the causative agents of CLABSI may be linked to their persistence in the environment, leading to exogenous transmission. These organisms can directly enter a patient’s body during insertions or manipulation of CVC through contamination from healthcare workers’ hands and environmental surfaces [61,62]. A systematic review has found that Gram-negative bacteria can survive longer than Gram-positive bacteria, hence contributing more towards healthcare-associated infections via contaminated sinks and water systems [63]. The lower rates of Gram-positive organisms are likely attributable to screening and targeted decolonization efforts for Staphylococcus aureus in ICUs [64]. MDRO is a growing problem in this region. According to the WHO’s Western Pacific Regional Office, Malaysia is projected to experience 87,000 deaths between 2020 and 2030 if no significant action is taken to address AMR [65]. The rise in MDR Gram-negative bacteria complicates clinical management and emphasizes that the urgent need for continuous surveillance, education of healthcare workers and strict monitoring with audit and feedback are essential to mitigate the burden of CLABSI and improve patient outcomes [66]. For instance, training programs such as the WHO-based Train-the-Trainers (TTT) courses have significantly improved knowledge scores post-training, with a 21.2% increase reported in Malaysia [67].

This study has several strengths. It provides valuable insights into CLABSI incidence, risk factors, and compliance with the CVC care bundle in Malaysian ICUs. The multi-site design, which includes three tertiary referral hospitals and one teaching hospital, improves the generalizability of the findings to similar healthcare settings across Malaysia. However, all participating centers were located within a single upper middle-income country. Therefore, the findings may not be directly generalizable to regions with different healthcare settings or resource availability. Further studies in diverse healthcare contexts are needed to validate the applicability of our results in other settings. Using standardized definitions, the study facilitates benchmarking within hospitals and internationally. Furthermore, it addresses a significant knowledge gap regarding the relationship between CLABSI incidence and care bundle compliance, a topic previously underexplored in this region. The study’s findings could be crucial for healthcare policymakers and infection-control teams in strengthening infection-prevention practices.

Furthermore, the detailed microbiological profile highlights the importance of antimicrobial stewardship programs and targeted infection-prevention and -control (IPC) strategies to prevent the spread of multidrug-resistant organisms. Despite these strengths, this study has several limitations that warrant consideration. Firstly, the relatively short study duration and low event rate constrained the statistical power, particularly for identifying trends and risk factors. Second, the reliance on paper-based systems in some participating hospitals posed challenges in data completeness. To address these limitations, future studies should conduct research over longer durations, larger sample sizes to enhance the robustness of statistical findings and adoption of standardized electronic data-capture systems to enhance data accuracy. Extending the duration of the study could provide more comprehensive data and allow for the assessment of long-term trends and the effectiveness of interventions over time.

## 5. Conclusions

This study reports a CLABSI rate of 4.16 per 1000 catheter days in Malaysian ICUs, with an average compliance rate of 65%. Multivariate analysis identified prolonged ICU stays, surgery, and the use of multiple catheters as key risk factors for CLABSI. Ongoing surveillance, targeted interventions and strong infection-prevention practices—supported by rigorous monitoring, audits and feedback—are essential to reducing CLABSI incidence and improving patient outcomes. The predominance of Gram-negative multidrug-resistant bacteria emphasizes the urgent need for targeted infection-prevention strategies and antimicrobial stewardship programs. Future research should focus on larger studies examining the impact of implementation strategies and the cost-effectiveness of interventions, and behavioral studies to further improve CLABSI-prevention efforts.

## Figures and Tables

**Figure 1 antibiotics-14-00271-f001:**
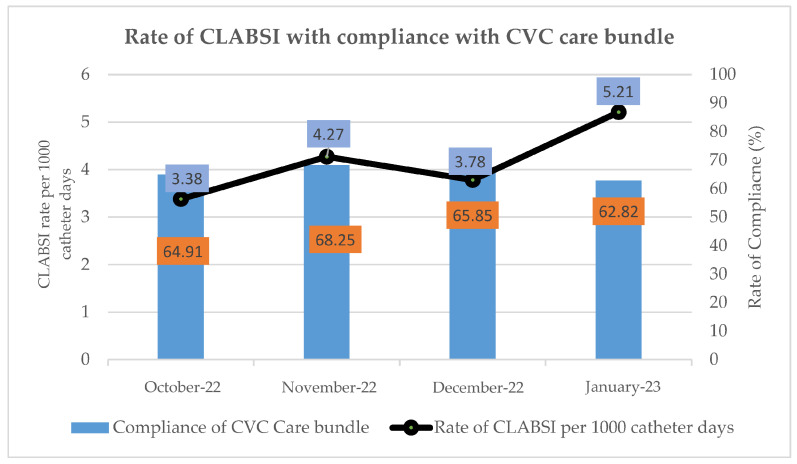
Overall rate of CLABSI with compliance of CVC care bundle in ICU of three hospitals.

**Table 1 antibiotics-14-00271-t001:** Patient demographic data and catheter details and outcome (N = 862).

Characteristic	n (%)
Gender	
Male	517 (60.0)
Female	345 (40.0)
Age (median, IQR) *	59.0 [40.0–69.0]
Duration of hospital stay (median, IQR) *	15.0 [8.0–46.0]
Duration of ICU stay (median, IQR) *	8.0 [3.0–16.0]
**Invasive Procedure or devices**	
Dialysis within 30 days	
Yes	120 (13.9)
No	742 (86.1)
Surgery within 30 days	
Yes	317 (37.0)
No	545 (63.0)
Mechanical ventilation	
Yes	813 (94.3)
No	49 (5.3)
Arterial line	
Yes	834 (96.8)
No	28 (3.2)
Peripheral line	
Yes	811 (94.0)
No	51 (5.9)
Urinary catheter/Foley	
Yes	832 (96.5)
No	30 (3.5)
Charlson comorbidity index (median, IQR)	4.0 [2.0–9.0]
APACHE II score (median, IQR)	19.0 [13.0–28.0]
Outcome of the patient *	
Transfer out to ward	680 (78.9)
Mortality in ICU	174 (20.2)
Discharge at own risk	8 (0.9)
**Catheter Details**	
Site of insertion	
Jugular vein	404 (39.3)
Femoral vein	595 (57.9)
Subclavian vein	28 (2.7)
Number of lumens	
Double	127 (12.4)
Triple/more	900 (87.6)
Lock solution	
Anticoagulant	151 (14.7)
No lock solution	876 (85.3)
Category of catheter insertion	
Elective	190 (18.5)
Emergency	561 (54.6)
Not documented	276 (26.7)
Catheter outcome	
Removed	791 (77.0)
CLABSI	18 (0.7)
Malfunction	60 (5.2)
No longer need	642 (55.6)
Other reason	71 (6.2)

* IQR = Interquartile range.

**Table 2 antibiotics-14-00271-t002:** Distribution of microorganisms’ distribution among CLABSI patients in ICU.

Items	n (%)
Total number of organisms	23
Total number of MDRO	12 (52.2)
Gram-negative bacteria	(n = 18, 78.3%)
*Acinetobacter baumanii*	2 (8.69)
*Burkholderia cenocepacia*	2 (8.69)
*Elizabethkingia anophelis*	1 (4.35)
*Escherichia coli*	1 (4.35)
*Enterobacter cloacae*	1 (4.35)
*Klebsiella pneumoniae*	7 (30.4)
*Pseudomonas aeruginosa*	1 (4.35)
*Stenotrophomonas maltophilia*	2 (8.69)
*Serratia marcescens*	1 (4.35)
Gram-positive bacteria	(n = 4, 17.4%)
*Enterococcus faecalis*	2 (8.69)
*Pseudomonas chlororaphis*	1 (4.35)
*Staphylococcus aureus*	1 (4.35)
Fungal	(n = 1, 4.35%)
*Candida dubliniensis*	1 (4.35)

**Table 3 antibiotics-14-00271-t003:** Regression analysis of significant risk factors for central line-associated bloodstream infection (CLABSI) in the univariate and multivariate analysis before and after propensity score matching.

Item	Before PS Matching				After PS Matching			
	Univariate Analysis OR (95% CI)	*p*-Value	Multivariate Logistics Regression AOR (95% CI)	*p*-Value	Univariate Analysis OR (95% CI)	*p*-Value	Multivariate Logistics Regression AOR (95% CI)	*p*-Value
Age (Med, IQR)	1.028 (0.997–1.060)	0.074			1.033 (0.993–1.075)	0.102	1.045 (0.969–1.127)	
Gender (Male)	1.008 (0.385–2.643)	0.987			1.406 (0.490–4.036)	0.525		
Duration of ICU stay (Med, IQR)	1.012 (1.000–1.023)	0.044	1.007 (0.996–1.019)	0.209	1.074 (1.030–1.120)	0.000	1.994 (1.092–3.009) **	0.001
Primary reason for admission								
Sepsis	1.189 (0.453–3.119)	0.725			0.891 (0.310–2.564)	0.830		
Shock	0.811 (0.262–2.506)	0.716			0.571 (0.170–1.925)	0.366		
Invasive procedures within 30 days								
Dialysis	4.356 (1.669–11.369)	0.003	1.668 (0.502–5.544)	0.404	0.515 (0.178–1.485)	0.219	1.791 (0.350–9.157)	0.453
Surgery	1.573 (0.610–4.055)	0.349			7.429 (2.208–24.995)	0.001	2.021 (1.468–5.830) *	0.023
Comorbidities								
Diabetes mellitus	1.033 (0.394–2.707)	0.948			1.486 (0.518–4.266)	0.461		
Hypertension	0.422 (0.156–1.142)	0.089	0.776 (0.237–2.547)	0.676	0.423 (0.143–1.251)	0.120	0.595 (0.133–2.679)	0.321
Dyslipidemia	0.513 (0.178–1.476)	0.216	0.598 (0.169–2.111)	0.424	0.371 (0.107–1.291)	0.119	0.257 (0.045–1.460)	0.133
Charlson comorbidity index (Med, IQR)	1.248 (1.047–1.487)	0.013	1.168 (0.906–1.504)	0.230	1.181 (0.953–1.453)	0.129	0.836 (0.588–1.187)	0.268
APACHE II score (Med, IQR)	1.072 (1.034–1.111)	0.00	1.052 (1.006–1.100)	0.025	1.002 (0.939–1.069)	0.957		
(B) Catheter details								
More than 1 catheter in situ	5.625 (2.142–14.768)	0.00	3.401 (1.034–11.180)	0.044	4.436 (1.433–13.731)	0.010	3.167 (1.519–9.313) ***	<0.001
Catheter site Jugular femoral	-	-			4.750 (1.597–14.126) 0.211 (0.071–0.626)	0.005	2.859 (2.414–4.339)	0.387
No. of lumen Two Three/more	0.672 (0.087–5.177) 1.487 (0.193–11.455)	0.703			0.365 (0.044–3.062) 2.742 (0.328–22.947)	0.334		
Use of lock solution Anticoagulant No lock solution	1.083 (0.243–4.839) 0.923 (0.207–4.123)	0.917			0.625 (0.127–3.081) 1.600 (0.325–7.888)	0.564		
Category of catheter insertion Elective Emergency	9.390 (1.857–47.482) 5.907 (1.280–27.268)	0.007 0.023	3.601 (0.183–70.881) 3.349 (0.167–67.232)	0.399 0.430	3.167 (0.564–17.778) 2.714 (0.538–13.583)	0.190 0.226		
Category of operator MO	0.709 (0.270–1.856)	0.483			0.280 (0.096–0.818)	0.027	2.227 (0.148–3.440)	0.172
USG catheter insertion	0.615 (0.198–1.917)	0.402			0.225 (0.065–0.778)	0.018	0.075 (0.003–1.907)	0.452
Hand hygiene	5.398 (1.544–18.869)	0.008	3.343 (0.265–44.488)	0.345	2.059 (0.539–7.860)	0.291		
Maximal barrier precaution	5.398 (1.544–18.869)	0.008	3.343 (0.265–44.488)	0.345	2.059 (0.539–7.860)	0.291		
Chlorhexidine antisepsis	5.398 (1.544–18.869)	0.008	3.343 (0.265–44.488)	0.345	2.059 (0.539–7.860)	0.291		
Overall compliance	0.585 (0.223–1.532)	0.275			1.545 (0.527–4.529)	0.428		
Mortality in ICU	0.785 (0.223–2.761)	0.706			0.558 (0.145–2.143)	0.395		

* *p* < 0.05, ** *p* < 0.01, *** *p* < 0.001. (After PS matching: Hosmer and Lemeshow test, c2(8) = 4.998, *p* = 0.758; Cox and Snell R2 = 0.354; Nagelkerke R2 = 0.560.) AOR, adjusted odds ratio; CI, confidence interval.

**Table 4 antibiotics-14-00271-t004:** Characteristics of the hospitals in this study.

Name of Hospital	Location of Hospital	Type of Hospital	Bed Capacity	No. of Wards	No. of ICUs	General ICU Beds	Type of Medical Record
Hospital Kuala Lumpur (HKL)	Kuala Lumpur	Tertiary referral	2300	83	5	30	Manual and electronic in ICU
Hospital Queen Elizabeth II (HQEII)	Kota Kinabalu, Sabah	Tertiary referral	400	18	3	16	Manual only
Universiti Malaya Medical Centre (UMMC)	Kuala Lumpur	Tertiary referral	1643	44	5	20	Electronic only

**Table 5 antibiotics-14-00271-t005:** Definition of study outcomes.

Item	Definition
Central line-associated bloodstream infection (CLABSI)	Laboratory-confirmed bloodstream infection (LCBI) where an eligible bloodstream infection (BSI) organism is identified and an eligible central line is in place for more than two calendar days on date of event, with day of device placement being Day 1.
Central line (CL)	An intravascular catheter that terminates at or close to the heart or in one of the great vessels and is used for infusion, withdrawal of blood or hemodynamic monitoring.
Eligible central line (CL)	Central lines (CLs) that have been in place for more than two consecutive calendar days (on or after CL Day 3), following the first access of the CL in an inpatient location during the current admission.
Eligible BSI organism	Any organism that is eligible for use in meeting LCBI or MBI-LCBI criteria. In other words, an organism that is not an excluded pathogen for use in meeting LCBI or MBI-LCBI criteria (https://view.officeapps.live.com/op/view.aspx?src=https%3A%2F%2Fwww.cdc.gov%2Fnhsn%2Fxls%2Fmaster-organism-com-commensals-lists.xlsx&wdOrigin=BROWSELINK, accessed on 3 March 2025)
Multidrug resistance organisms (MDROs)	Non-susceptible to at least one agent in three or more designated antimicrobial categories [42]

## Data Availability

The dataset used in this study is not publicly available due to institutional restrictions. However, all relevant data supporting the findings of this study have been included in the article. Further inquiries can be directed to the corresponding author.

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
