# Peer review of "Incidence, Compliance, and Risk Factor Associated with Central Line-Associated Bloodstream Infection (CLABSI) in Intensive Care Unit (ICU) Patients: A Multicenter Study in an Upper Middle-Income Country"

_antibiotics, 2025, doi:10.3390/antibiotics14030271_

Round 1

Reviewer 1 Report

Comments and Suggestions for Authors

introduction

define what care bundles are

2.4 research instruments

line 134 specify how you introduce care bundles into a clinical practice

3.2 clabsi rates and compliance cvc  care bundles

specify  which bundles have least adherence this could help to understand how  can improve infection and prevention target measure 

3.4 clabsi risk factors

i would add if there is a colonization with MDRO, particularly colonization dected through rectal swab of carbapenemase-producing gram negative bacteria

4. discussion :

line 388 considering that the most of CVCs are in femoral site , i would also consider the insertion site as a risk factors for the develpment of clabsi caused by gram negative bacteria

Author Response

Comment 1: introduction-define what care bundles are
Response : Thank you for your feedback. We have incorporated the requested sentences in lines 94-107 of the revised manuscript.These strategies are often implemented through a care bundle approach, which consists of packages of at least three to five evidence-based strategies grouped together to maximize their effectiveness and optimize patient care (Lavallée et al., 2017). Care bundles are widely used in healthcare settings to prevent and manage various health conditions such as in preventing ventilators associated with pneumonia, surgical site infection (SSI) and pressure ulcer (Allegranzi et al., 2016; Lavallee et al., 2017). Insertion and maintenance of central venous access device bundles demonstrate reductions in the frequency of complication and bloodstream infection when implemented with compliance monitoring (Blot et al., 2014; Ista et al., 2016; Zingg & Pittet, 2016).  In Malaysia, the Ministry of Health (MOH) introduced a CVC care bundle guideline which included a checklist in 2016 aligning local practices with international standards and to address this ongoing challenge (CDC/NHSN 2022, MOH 2016). The components of the CVC care bundle include the following practices: 1) hand hygiene; 2) maximal barrier precautions; 3) chlorhexidine skin antisepsis; 4) optimal site selection and 5) daily review of (CVC) line (MOH, 2016).

Comment 2: 
2.4 research instruments-line 134 specify how you introduce care bundles into a clinical practice
Response: Thank you for your question. We have incorporated the requested sentences in lines 149–152 of the revised manuscript. The CVC Care Bundle Guidelines by the Ministry of Health (MOH) Malaysia were published in 2016. All healthcare facilities are required to comply with these guidelines as part of a national initiative to standardize the use of CVCs in order to reduce CLABSI (MOH, 2016).
Additionally, we have added the following statement in the line 168-173 of the revised manuscript.
Prior to data entry, the primary investigators and two research assistants from each participating hospital were trained in data extraction by the research coordinator us-ing a standardized training module based on the surveillance protocol. This training included CLABSI identification and definition, as well as compliance assessment with the CVC Care Bundle to ensure consistency and accuracy in data collection across all study sites.

Comment 3: 
3.2 clabsi rates and compliance cvc  care bundles -specify  which bundles have least adherence this could help to understand how  can improve infection and prevention target measure 
Response: Thank you for your question. We have addressed this in the manuscript in the line 213-216. Based on the data in Table 6 (Supplementary File), adherence rates for four compo-nents of the CVC Care Bundle were as follows: hand hygiene (69.5%), maximal barrier precautions (69%), chlorhexidine antisepsis (66.9%), and daily review (77.1%). Among these, chlorhexidine antisepsis had the lowest adherence.

Comment 4: 3.4 clabsi risk factors-i would add if there is a colonization with MDRO, particularly colonization dected through rectal swab of carbapenemase-producing gram negative bacteria
Response: Thank you for your suggestion. However, our study did not collect data on MDRO colonization, including colonization detected through rectal swabs for carbapenemase-producing Gram-negative bacteria. Our data collection focused on microbiological information related to infections associated with CVCs rather than colonization surveillance.

Comment 5: 4. discussion -line 388 considering that the most of CVCs are in femoral site , i would also consider the insertion site as a risk factors for the develpment of clabsi caused by gram negative bacteria
Response: Thank you for highlighting this point. In our study, site of the insertion was one of  the variables in the risk factor analysis and not  found to be a statistically significant risk risk factor for CLABSI. While previous studies have suggested an association, our findings did not demonstrate a significant correlation in our study population.

Reviewer 2 Report

Comments and Suggestions for Authors

The topic is highly relevant to current healthcare challenges, particularly in ICU settings where CLABSIs significantly affect patient outcomes and healthcare costs. The prospective observational study design employed across multiple centers enhances the reliability and generalizability of the findings. The inclusion and exclusion criteria are clearly defined, which aids in understanding the study population. Extensive data on patient demographics, clinical details, and compliance with the CVC Care Bundle were systematically collected and analyzed.

The study reveals a compliance rate to the CVC Care Bundle that varies significantly across the centers. This variation could introduce biases in the incidence rates reported and may not accurately reflect the effectiveness of compliance on CLABSI rates.

While the study includes multiple centers, all are located within a single upper middle-income country. This may limit the applicability of the findings to regions with different healthcare settings or resource availability.

The study covers only a four-month period, which may not capture seasonal variations in CLABSI rates or the long-term effects of interventions.

Suggestions: Extending the duration of the study could provide more comprehensive data and allow for the assessment of long-term trends and the effectiveness of interventions over time.

Strategies to improve adherence to the CVC Care Bundle across all centers could be strengthened to provide more uniform data and potentially reduce CLABSI rates.

The discussion could be expanded to more critically address the limitations of the study, particularly the potential biases introduced by varying compliance rates and the implications of these for the study's conclusions.

Comments on the Quality of English Language

The topic is highly relevant to current healthcare challenges, particularly in ICU settings where CLABSIs significantly affect patient outcomes and healthcare costs. The prospective observational study design employed across multiple centers enhances the reliability and generalizability of the findings. The inclusion and exclusion criteria are clearly defined, which aids in understanding the study population. Extensive data on patient demographics, clinical details, and compliance with the CVC Care Bundle were systematically collected and analyzed.

The study reveals a compliance rate to the CVC Care Bundle that varies significantly across the centers. This variation could introduce biases in the incidence rates reported and may not accurately reflect the effectiveness of compliance on CLABSI rates.

While the study includes multiple centers, all are located within a single upper middle-income country. This may limit the applicability of the findings to regions with different healthcare settings or resource availability.

The study covers only a four-month period, which may not capture seasonal variations in CLABSI rates or the long-term effects of interventions.

Suggestions: Extending the duration of the study could provide more comprehensive data and allow for the assessment of long-term trends and the effectiveness of interventions over time.

Strategies to improve adherence to the CVC Care Bundle across all centers could be strengthened to provide more uniform data and potentially reduce CLABSI rates.

The discussion could be expanded to more critically address the limitations of the study, particularly the potential biases introduced by varying compliance rates and the implications of these for the study's conclusions.

Author Response

Comment 1: The study reveals a compliance rate to the CVC Care Bundle that varies significantly across the centers. This variation could introduce biases in the incidence rates reported and may not accurately reflect the effectiveness of compliance on CLABSI rates.
Response: We appreciate the reviewer’s comment regarding the variation in compliance rates across centers and its potential impact on the reported CLABSI incidence rates. However, we would like to clarify that our manuscript does not present compliance rates at the individual hospital level. Instead, we report overall compliance rates across all study sites. Due to the small sample size, presenting center-specific compliance rates would result in underpowered analyses. If the reviewer is referring to potential unmeasured variations, we acknowledge that differences in compliance at the hospital level could influence CLABSI rates. However, our analysis was based on aggregated compliance data rather than center-specific rates, which we believe provides a more comprehensive assessment of overall adherence and CLABSI outcomes.

Comment 2: While the study includes multiple centers, all are located within a single upper middle-income country. This may limit the applicability of the findings to regions with different healthcare settings or resource availability.
Response: Thank you for your thoughtful comment. We acknowledge that the four-month study duration may not fully capture seasonal variations in CLABSI rates or the long-term effects of the intervention. As noted in our manuscript, the relatively short duration and low event rate constrained the statistical power, particularly for identifying trends and risk factors. To address these limitations, we have already highlighted the need for future studies to be conducted over longer durations and with larger sample sizes to enhance the robustness of statistical findings. We appreciate the reviewer’s suggestion, and we believe our existing discussion aligns with this recommendation. However, we have further emphasize this point in the revised manuscript as suggested by the reviewer and added the following comment in line 402-404 under the limitation section.

Comment 3: 

The study covers only a four-month period, which may not capture seasonal variations in CLABSI rates or the long-term effects of interventions. Suggestions: Extending the duration of the study could provide more comprehensive data and allow for the assessment of long-term trends and the effectiveness of interventions over time.
Response: Thank you for your thoughtful comment. We acknowledge that the four-month study duration may not fully capture seasonal variations in CLABSI rates or the long-term effects of the intervention. As noted in our manuscript, the relatively short duration and low event rate constrained the statistical power, particularly for identifying trends and risk factors. To address these limitations, we have already highlighted the need for future studies to be conducted over longer durations and with larger sample sizes to enhance the robustness of statistical findings. We appreciate the reviewer’s suggestion, and we believe our existing discussion aligns with this recommendation. However, we have further emphasize this point in the revised manuscript as suggested by the reviewer and added the following comment in line 402-404 under the limitation section.

Comment 4: Strategies to improve adherence to the CVC Care Bundle across all centers could be strengthened to provide more uniform data and potentially reduce CLABSI rates.
Response: Thank you for your suggestion. We agree that strengthening strategies to improve adherence to the CVC Care Bundle across all centers could enhance data uniformity and potentially reduce CLABSI rates. We have included this under the Discussion section under the line 318-320.Hence, potential interventions such as standardized training, routine audits and feedback mechanisms to reinforce adherence are essential to sustain high compliance rates and effectively reduce CLABSI incidence.

Comment 5: The discussion could be expanded to more critically address the limitations of the study, particularly the potential biases introduced by varying compliance rates and the implications of these for the study's conclusions.
Response: Thank you for your valuable feedback. We acknowledge that varying compliance rates across centers could introduce potential biases and impact the study’s conclusions. However, due to the low incidence rates of CLABSI in individual centers (HQEII: 2 cases; UMMC and HKL: 8 cases each), we opted to analyze the data as a whole rather than by individual centers to ensure statistical robustness.

Round 2

Reviewer 2 Report

Comments and Suggestions for Authors

The manuscript has been sufficiently improved.